# Monocyte-to-Lymphocyte Ratio as a Predictor of Worse Long-Term Survival after Off-Pump Surgical Revascularization-Initial Report

**DOI:** 10.3390/medicina57121324

**Published:** 2021-12-03

**Authors:** Tomasz Urbanowicz, Michał Michalak, Anna Olasińska-Wiśniewska, Anna Witkowska, Michał Rodzki, Ewelina Błażejowska, Aleksandra Gąsecka, Bartłomiej Perek, Marek Jemielity

**Affiliations:** 1Cardiac Surgery and Transplantology Department, Poznan University of Medical Sciences, 61-701 Poznan, Poland; anna.olasinska@poczta.onet.pl (A.O.-W.); anna.witkowska@skpp.edu.pl (A.W.); michal.rodzki@skpp.edu.pl (M.R.); bperek@ump.edu.pl (B.P.); mjemielity@poczta.onet.pl (M.J.); 2Department of Computer Science and Statistics, Poznan University of Medical Sciences, 61-701 Poznan, Poland; michal@ump.edu.pl; 31st Chair and Department of Cardiology, Medical University of Warsaw, 02-091 Warsaw, Poland; eb.blazejowska@gmail.com (E.B.); gaseckaa@gmail.com (A.G.)

**Keywords:** OPCAB, MLR, inflammatory response, survival

## Abstract

*Background and objective:* Coronary artery disease is one of the leading causes of deaths nowadays and the trends in diagnosis and revascularization are still in plateau despite well-known factors. Simple whole blood count parameters may be used to measure inflammatory reactions that are involved in processes of atherosclerosis progression. The aim of our study was to analyse the association between simply available hematologic indices and long-term mortality following off-pump coronary artery bypass grafting (OPCAB). *Material and Methods:* The study group comprised 129 consecutive patients (16 females and 113 males, mean age 66 ± 6 years) who underwent surgical revascularization with off-pump technique between January 2014 and September 2019. The mean follow-up was 4.7 +/−1.9 years. A receiver operating characteristics curve was applied to estimate demographical and perioperative parameters including MLR for mortality. *Results:* Cox regression analysis revealed chronic pulmonary obstructive disease (HR = 2.86, 95%CI 1.05–7.78), MLR (HR = 3.81, 95%CI 1.45–10.06) and right coronary artery blood flow (HR = 1.06, 95%CI 1.00–1.10) as significant factors predicting increased mortality risk. In the presented model, the MLR > 1.44 on 1st postoperative day was a significant predictor of late mortality after the OPCAB procedure (HR = 3.82, 95%CI 1.45–10.06). *Conclusions:* Pronounced inflammatory reaction after off-pump surgery measured by MLR > 1.44 can be regarded as a worse long-term prognostic factor.

## 1. Introduction

Coronary artery disease is the leading cause of death causes nowadays and the trends in diagnosis and revascularization are still in plateaued despite well-known risk factors [1]. CLARIFY registry presents the high possibility for interventional treatment in a wide group of patients with chronic coronary syndrome despite evidence-based applied therapies [2]. Surgical revascularization is still a method of choice for treatment of complex coronary disease with established safety and efficacy [3].

Atherosclerosis progression is the major feature of chronic coronary syndromes as the processes of plaque growth are related to inflammation, thrombosis, lipid disturbances, vascular smooth cell activation, remodeling, thrombosis, platelet activation and genetic factors [4]. Chronic inflammation in humans is associated with accelerated development of cardiometabolic diseases; in turn, the blocking of the IL-6 signaling pathway was associated with reduced cardiovascular event rates, independent of lipid-lowering in CANTOS study (The Canakinumab Anti-inflammatory Thrombosis Outcome Study) [5]. Ji et al., in their retrospective analysis, found monocyte to lymphocyte counts as an independent risk factor of the coronary artery disease presence and lesions’ severity [6].

The inflammatory background of atherosclerotic lesions development and progression is enhanced by interventional procedures that reflect redox imbalance. Surgical revascularization performed in an off-pump technique called Off-pump coronary artery bypass grafting (OPCAB) allows for the reduction of postoperative complications including systematic inflammatory response, myocardial, cerebral, and renal injury as compared to on-pump coronary artery bypass grafting [7]. Though the inflammatory response is reduced in off-pump compared to on-pump surgery, still inflammatory processes activation after off-pump procedures has been observed [8,9].

The molecular and cellular basis of percutaneous interventions’ failure results from inflammatory overreaction [10,11]. Several approaches to limit inflammatory reactions and prevent atherosclerosis progression were already postulated [12,13,14].

The aim of our study was to analyse if easily accessible haematological indices representing inflammatory processes activation may be associated with long-term survival after surgical revascularization.

## 2. Patients and Methods

### 2.1. Demographical and Clinical Characteristics

The study group comprised 129 consecutive patients (16 females and 113 males) with the mean age of 66 ± 6 who underwent OPCAB procedure between January 2014 and September 2017 in our hospital. Only those subjects who had grafts blood flow measurements performed were included from 856 patients operated with off-pump technique in our department. All patients were qualified for the OPCAB procedure after careful analysis of clinical history and examinations’ results by the Heart Team. Patients included into the study were diagnosed with chronic coronary syndrome; subjects with unstable angina and acute non-ST and ST myocardial infarction were excluded from the analysis. The indication for surgery included: left main stem stenosis in 103 (80%) patients followed by two vessels disease in 18 (14%) and three vessels disease in 8 (6%) patients. The perioperative mortality risk was assessed by Euroscore II and presented in Table 1.

The study group was retrospectively divided into two subsets based on survival results within mean observation time of 4.7 +/− 1.9 years. The baseline characteristics are presented in Table 1.

### 2.2. Laboratory Tests

Laboratory results were obtained on admission, including whole blood count, Troponin I serum levels, creatinine, and fibrinogen. Any clinical signs of infection of chronic inflammatory processes or oncologic history were disqualified from the study. The whole blood count was the only standard laboratory test presenting inflammatory response. The other inflammatory parameters, such as C-reactive protein (CRP) or procalcitonin, were not routinely assessed. NLR and MLR were calculated as the ratio of neutrophils to lymphocyte and monocyte to lymphocyte counts, respectively (Table 1).

The study was approved by the Institutional Ethics Committee (Protocol No 55/20 from 16 January 2020) and respected the principles outlined in the Declaration of Helsinki.

### 2.3. OPCAB Procedures

All procedures were performed by the same experienced team in general anaesthesia through median sternotomy. The procedures were performed after full heparinization under activated clotting time (ACT) (457 +/− 32 min) time. The anastomoses were formed with intraluminal shunts application, and the operation field was stabilized by Octopus III (Medtronic, USA). After a second dose of protamine, the graft blood flows were measured. Detailed information is presented in Table 2.

### 2.4. Post-Discharge Mortality

All patients were followed up in the outpatient clinic. Overall mortality was presented as a percentage of patients. All deceased cases were easily identified since we have limited access (confined only to the treated patients) to the National Health Care System and presented as all-cause mortality.

### 2.5. Statistical Analysis

Continuous data were presented as means and standard deviations or medians and interquartile ranges [Q1–Q3] if data did not follow the normal distribution (Shapiro–Wilk test). Nominal data were presented as numbers and percentages. The comparison between survival and deceased patients was performed with the use of Mann–Whitney test or Chi-square test for independence. The ROC analysis was performed in order to find an optimal cut-off point for potential prognostic parameters of deceased patients. A Cox regression model was performed to assess the risk of death for analyzed parameters. The analysis was performed twice, once as univariable and second as a multivariable model. The multivariable model was assessed by performing a stepwise backward selection procedure. The results were presented as hazard ratios (HR) and their 95%CI. A Kaplan–Meier estimate was used to present the survival rate. The comparison of survival curves was performed with the use of a Log-rank test.

The calculations were performed using the statistical package MedCalc^®^ Statistical Software version 19.6 (MedCalc Software Ltd., Ostend, Belgium; https://www.medcalc.org, accessed on 27 October 2021). All tests were considered significant at *p* < 0.05.

## 3. Results

A total of 129 (113 (88%) males and 16 (12%) females) patients were referred for surgical revascularization due to complex chronic coronary disease. There were no early deaths in the presented group and 17% (22 patients) long-term mortality in a mean observation time of 4.7 +/− 1.9 years. The mean number of performed grafts was 1.99, including 129 (50%) left internal mammary arteries (LIMA), 40 (16%) right internal mammary artery (RIMA) and 86 (34%) venous grafts. All operations were performed in off-pump technique and the grafts’ blood flows were measured as included in Table 2. The laboratory results obtained after the first 24 h are presented in Table 2.

The Mann–Whitney test for independent samples revealed a significant difference in MLR obtained on the 1st postoperative day between groups (*p* = 0.037). The receiver operator curve (ROC curve) shows that MLR has prognostic properties for survival AUC = 0.66, *p*-value 0.037 with 58.8% sensitivity and 75% specificity as presented in Figure 1.

Abbreviations: AUC—area under the curve, MLR—monocyte to lymphocyte ratio.

The Mann–Whitney test revealed a significant difference in right coronary artery blood flow measurement between survival and deceased patients (*p* = 0.0001). There was no significant difference in left descending artery (*p* = 0.909) and circumflex artery blood flow (*p* = 0.973) between patients. The ROC analysis for RCA blood flow shows predictive properties for survival (AUC = 0.81, *p* < 0.001), giving sensitivity 100% and specificity 67.64%, as presented in Figure 2a–c.

Abbreviations: AUC—area under the curve, LAD—left descending artery, Cx—circumflex artery, RCA—right coronary artery.

Among comorbidities, cox regression analysis revealed chronic pulmonary obstructive disease (COPD) as a significant factor predicting increased mortality risk HR = 2.86, 95%CI 1.06–7.77 (*p* = 0.039). Only the right coronary artery blood flow revealed significance for long-term survival in Cox regression analysis HR = 1.06, 95%CI 1.01–1.11 (*p* = 0.038). Detailed data of univariable analysis are presented in Table 3.

Cox regression analysis revealed MLR value above 1.44 on the first postoperative day as significant for predicting increased mortality risk (HR = 3.82, 95%CI 1.45–10.06) as presented in Table 3.

Postoperative follow-up was supported by outpatients’ regular visits. The survivors’ group is clinically free from anginal symptoms.

In the multivariable analysis, only MLR > 1.44 was an independent predictor of long-term mortality. Patients with MLR 1-st day > 1.44 have a five-year survival rate of 83.8%, which is significantly lower compared to patients whose monocyte to lymphocyte ratio ≤ 1.44, with the five-year survival rate of 96.6% (*p* = 0.0037) as presented in Figure 3.

Abbreviations: MLR—monocyte to lymphocyte ratio.

The mean survival time was 7.0 years; 95%CI (96.8–7.2) and 6.3 years; 95%CI (5.8–6.8) for patients with MLR 1st ≤ 1.44 and MLR 1st > 1.44, respectively.

## 4. Discussion

The results of our study present the correlation between immediate postoperative inflammatory response and increased mortality rate in long-term observation. The monocyte to lymphocyte counts as a simple predictor for inflammatory overactivation was significantly higher in deceased patients in univariable analysis (1.5 (1.1–2) vs. 1.0 (0.8–1.5), *p* = 0.038). In our analysis, MLR over 1.44 was the only prognostic significant factor for survival.

The correlation between inflammatory reaction and coronary artery disease is postulated for both chronic and acute episodes [5,6]. The individual inflammatory response can be regarded as a key issue for prognosis. We present the results of the study, including patients who underwent the surgical procedure uneventfully but are characterized by different inflammatory responses to surgical intervention and prognosis in long-term follow-up. The different approaches to limit inflammatory reactions and prevent atherosclerosis in chronic coronary syndromes progression were already postulated [12,13,14].

Minimalization of inflammatory response in off-pump procedures was already presented in a low-volume Hazama et al. study [15]. The comparisons of oxidate stress levels between on-pump and off-pump revascularization procedures were performed on a group of 106 patients by Vukicevic et al. [16]. In his review, Borgerman postulated that inflammatory mediator system and cellular activation do not sufficiently and simply relate and explain the clinical course after surgical approach with cardiopulmonary bypass application but are more relevant to individual response based on genetic factors [17]. Zhai et al., in their study, performed in 123 patients, applied dexmedetomidine during off-pump procedure and achieved a reduction of pro-inflammatory factors levels (interleukin (IL)-6, tumor necrosis factor-α), resulting in amelioration of hemodynamic parameters [18]. The relation between increased neutrophil to lymphocyte (NLR) ratio in immediate postoperative time and mortality in off-pump patients was presented recently by Sahin & Sisli [19].

In our analysis, the Troponin–I levels did not influence survival rate; however, there were no patients with high troponin release and peri-operative myocardial infarction. We observed postoperative maximum median (Q1–Q3) serum values of 1 (0.6–3.05) mcg/L vs. 0.6 (0.4–3.3) mcg/L (*p* = 0.248), in the survivors and deceased groups, respectively. There was no significant difference in preoperative and postoperative left ventricular ejection fraction among both groups: 60% (53%–60%) vs. 60% (60%–60%) (*p* = 0.2473) and 60% (45%–60%) vs. 60% (50–60%) (*p* = 0.1729) in the survivors and deceased groups, respectively. Our results confirm the Cuculi et al. study performed on 31 patients referred for surgical revascularization. The authors concluded that perioperative systemic inflammation neither triggers myocardial injury nor depresses left ventricular ejection fraction [20].

This is the first, to the best of our knowledge, study presenting the relation between MLR as hematologic indices parameter and survival for off-pump procedures in long-term follow-up. The relation between MLR and venous graft failure was previously reported by Okzus et al. [21].

The prognostic value of inflammatory indices as MLR was found after percutaneous intervention in STEMI patients by Cai et al. in a retrospective analysis [22]. The meta-analysis by Quan et al. presented the relation between lymphocyte-to-monocyte ratio (LMR) and higher short-term and long-term mortality, especially for younger patients [23]. Retrospective analysis of 426 patients with non-ST elevation myocardial infarction (NSTEMI) patients revealed that LMR can be regarded as a biomarker for identifying high-risk individuals for the development of slow flow/no reflow phenomenon [24].

Our relatively small group of consecutive patients included in the study was operated on without any major complications. The inclusion criteria included blood flow measurements that are not routinely performed in our institution. All patients uneventfully underwent surgery without any episodes of acute kidney injury, infection, and perioperative ischemia/myocardial infarction. We believe that the results of this study point out the new perspective for patients undergoing surgical revascularization. The inflammatory overreaction in particular patients that may be noted by MLR increase may help in long-term follow-up. All patients underwent surgical coronary revascularization with an uneventful post-surgical period; however, those with early inflammatory overreaction after surgery presented worse long-term outcomes. The inflammatory reaction was characterized by an MLR increase with a cut-off value of 1.44.

Interestingly, the blood flow measurement in the left descending artery was not found to be a survival predictor in a mean observation time of 4.7 +/− 1.9 years. We believe that the presented results are too short of revealing dependency of left descending artery blood flow disturbances. The arterial grafts that were used including left internal mammary artery (n = 126) and right internal mammary (n = 40) are characterized by long-term patency rates. The results from Goldman et al. performed on VA cooperative study presented very good 10-year patency rates [25]. Therefore, we believe that our findings result from too short an observation period to reveal potential dependency of left descending artery blood flow disturbances. Conversely, we obtained significant results regarded right coronary artery blood flow. The right coronary artery was revascularized in 38 patients with venous grafts. While venous graft patency is lower than arterial ones, this finding supports our aforementioned opinion.

There is a scarcity of literature regarding monocyte-to-lymphocyte ratio and its relation to surgical coronary revascularization, especially regarding off-pump technique. The relation between monocyte to lymphocyte ratio and savenous graft patiency in surgical revascularization performed with on-pump technique on 89 patients was presented in Aydin et al. analysis [26]. Both the lymphocyte to monocyte ratio and neutrophil to lymphocyte ratio were independent predictors of saphenous vein graft disease and mortality [27]. Similarly, Okzus et al., in their analysis, found the link between inflammatory parameters including C-reactive protein (CRP) and lymphocyte-to-monocyte ratio and savenous grafts disease in patients undergoing on-pump surgery [21]. The relation between monocytes counts was also investigated in non-cardiac procedures, including peripherial vascular diseases suggesting a relation between limb ischemia and increased monocytes values [28]. The association between blood–brain barrier (BBB) disruption in acute ischemic stroke and monocytes counts were presented in the retrospective study, including 33 patients in Nadareishvili et al. analysis [29]. The MLR was also investigated in patients undergoing percutaneous coronary interventions (PCI) as possible future major acute coronary events (MACE) predictor [30]. Interestingly, in patients undergoing PCI due to non-ST myocardial infarction, MLR was regarded to be a useful biomarker for identifying slow flow/no flow phenomenon [24]. Presented results indicate not only long-term predictive significance but also an immediate link to hemodynamic disturbances related to inflammatory reactions. Fan et al., in their analysis, postulated a combination of MLR and NLR (neutrophil-to-lymphocyte ratio) as a possible prognostic valuable factor for long-term MACE risk in PCI patients [31].

Moreover, Karauzum et al., in their retrospective study, showed the relation between MLR and risk for contrast-induced nephropathy in acute coronary syndromes suggesting a relation between inflammatory parameters and peripheral blood perfusion disturbances [32]. Mentioned publications revealed the links between coronary and peripheral organs blood disturbances related to inflammatory markers such as monocytes. Cardiometabolic risk in relation to monocytes and serum cholesterol profiles was postulated by Sucato et al. [33]. MLR was also found as an independent predictor for postoperative cognitive dysfunction (POCD) following cardiovascular surgery, suggesting linkage to possible inflammatory processes [34]. Zeynalova et al., in their study, presented the relation between monocytes subtypes measured by flow cytometry and the 10-year risk of developing myocardial infarction and mortality following coronary heart disease [35]. Regarding cardiovascular diseases, possible prognostic values of MLR in pulmonary embolism should be mentioned and require further investigations [36,37].

Several inflammatory markers have already been investigated regarding the severity of coronary artery disease [38,39,40]. In their study, Tanriverdi et al. presented the C-reactive protein to albumin ratio (CAR) as an independent predictor of severity of disease [41]. Moreover, epicardial adipose tissue (EAT) was recently postulated as a novel factor recognized in atherosclerosis progression [42]. Atherosclerotic plaques not only promote inflammation but also trigger the EAT’s cytokines production, which eventually increases coronary artery disease severity. Chen et al. presented the expression of inflammatory factors and oxidative stress markers with correlation to calcium score in coronary artery disease in their publication [27].

### Study Limitation

This is a single-center retrospective study involving patients referred with a diagnosis of a stable coronary syndrome referred for coronary artery revascularization surgery. The group of 129 patients is relatively small and includes all-cause mortality. Differences in the numbers of female and male patients result, however, from the fact that it is a real-world study with the inclusion of consecutive all-comers. It requires further research in a larger population. Moreover, a longer observation time is needed to reveal the potential influence of blood flow in particular revascularized arteries.

## 5. Conclusions

The pronounced inflammatory reaction after off-pump surgery measured by MLR > 1.44 can be regarded as a worse long-term prognostic factor.

## Figures and Tables

**Figure 1 medicina-57-01324-f001:**
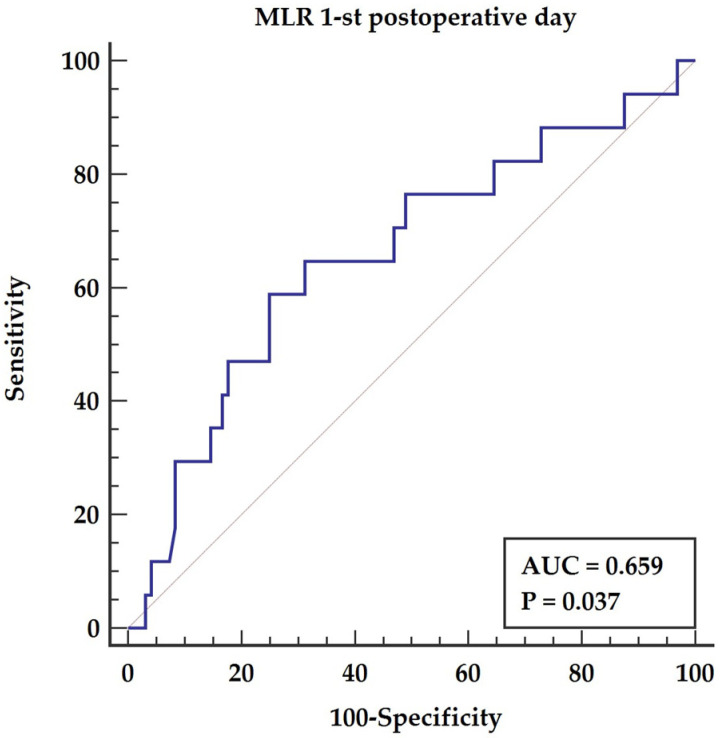
MLR in 1st postoperative day and survival probability.

**Figure 2 medicina-57-01324-f002:**
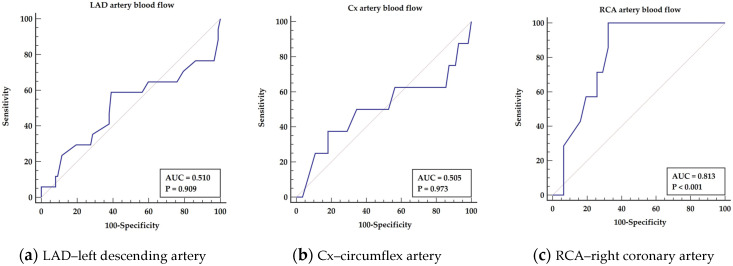
Blood flow measurements into coronary arteries.

**Figure 3 medicina-57-01324-f003:**
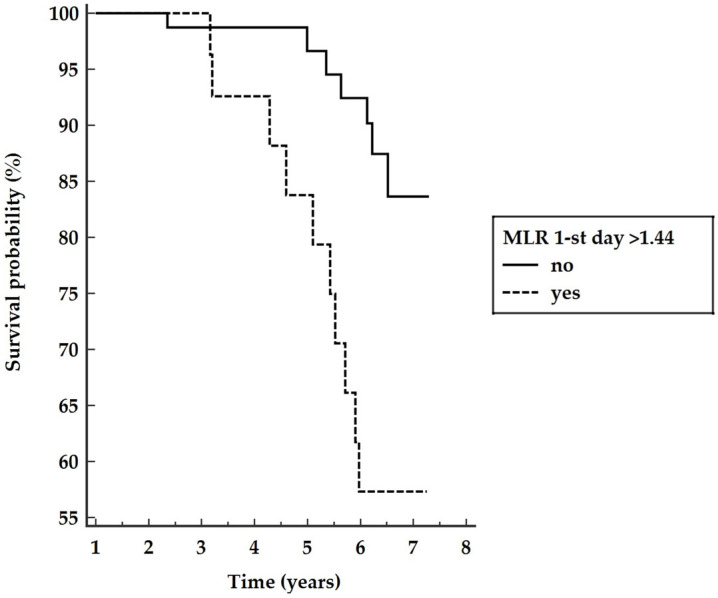
Kaplan–Meier survival curves for patients with MLR 1st above or below 1.44.

**Table 1 medicina-57-01324-t001:** Demographical, clinical and laboratory characteristics.

Parameter	Group 1 (Survivors) No = 107	Group 2 (Deceased) N = 22	*p*-Value
Age (median (Q1–Q3)	65 (58–70)	68 (63–74)	0.061
Gender (M (%)/K (%))	94 (88%)/13 (12%)	19 (86%)/3 (14%)	0.795
Coronary disease:			
1. Left main stem	89 (83%)	14 (64%)	0.043
2. Two vessels disease	12 (11%)	6 (27%)	0.047
3. Three vessels disease	6 (6%)	2 (10%)	0.493
Euroscore II; median (Q1–Q3)	1.5 (1.2–2.2)%	1.6 (1.3–2.1)%	0.548
Concomitant diseases:			
1. Arterial hypertension (%)	62 (58%)	9 (41%)	0.144
2. Hypercholesterolemia (%)	15 (14%)	2 (11%)	0.707
3. COPD (%)	6 (9%)	5 (23%)	0.059
4. Diabetes mellitus (%)	28 (26%)	7 (32%)	0.563
5. stroke (%)	7 (7%)	1 (5%)	0.731
6. kidney dysfunction (%)	7 (7%)	4 (18%)	0.098
Echocardiography:			
1. LVDd; median (Q1–Q3)	46 (41–52)	5 (44–55)	0.066
2. RVDd; median (Q1–Q3)	28 (26–31)	28 (25–32)	0.996
3. LAd; median (Q1–Q3)	37 (33–41)	4 (34–45)	0.232
4. IVSs; median (Q1–Q3)	13 (11–14)	14 (13–16)	0.049
5. LVEF; median (Q1–Q3)	60 (53–60)	60 (45–60)	0.090
Preoperative laboratory-morphology:			
1. WBC (median (Q1–Q3)	8 (6.9–9.4)	8.3 (7.2–9.4)	0.758
2. Neutrophils (median (Q1–Q3)	5.2 (4.2–6.2)	5 (4.8–6)	0.631
3. Lymphocyte (median (Q1–Q3)	2 (1.5–2.4)	1.9 (1.4–2.4)	0.588
4. Monocyte (median (Q1–Q3)	0.5 (0.4–0.6)	0.5 (0.4–0.6)	0.960
5. Hb (median (Q1–Q3)	8.9 (8.3–9.4)	8.5 (8–9.1)	0.114
6. Hct (median (Q1–Q3)	0.41 (0.4–0.44)	0.41 (0.38–0.43)	0.321
7. MLR (median (Q1–Q3)	0.3 (0.2–0.3)	0.3 (0.2–0.4)	0.417
8. NLR (median (Q1–Q3)	2.6 (2.1–3.3)	2.8 (2.1–3.7)	0.512
9. Plt (median (Q1–Q3)	229 (198–267)	243 (207–278)	0.554
Preoperative parameters:			
1. Creatinine (median (Q1–Q3)	84 (75–97)	87 (78–105)	0.465
2. Fibrinogen (median (Q1–Q3)	298 (264–358)	311 (248–378)	1.000
3. Troponin (median (Q1–Q3)	0 (0–0.03)	0 (0–0.03)	0.234

Abbreviations: Hct—hematocrit, Hb—hemoglobin, IVS—intraventricular septum diameter, LAd—left atrium diameter, LVEF—left ventricle ejection fraction, LVDd—left ventricle diastolic diameter, MLR—monocyte to lymphocyte ratio, NLR—neutrophil to lymphocyte ratio, Plt—platelets, RVDd—right ventricle diastolic diameter, WBC—white blood count.

**Table 2 medicina-57-01324-t002:** Intraoperative parameters and postoperative laboratory results (24 h).

	Group 1 (Survivors) No = 107	Group 2 (Deceased) No = 22	*p*
Number of grafts (mean):	2	1.86	
Applied grafts:			
1. LIMA (%)	107 (100%)	22 (100%)	-
2. RIMA (%)	33 (31%)	7 (32%)	0.926
3. SVBG (%)	74 (69%)	12 (55%)	0.204
Blood flow to:			
1. LAD mL/min (median (Q1–Q3)	15 (9–24)	18 (7–30)	0.898
2. Cx mL/min (median (Q1–Q3)	19 (12–29)	17 (9–34)	0.975
3. RCA ml/min (median (Q1–Q3)	30 (22–38)	45 (35–53)	0.010
Postoperative laboratory-morphology (24 h):			
1. WBC x109/L (median (Q1–Q3)	14.5 (12.5–16.8)	14.5 (12.5–16.8)	0.515
2. Neutrophils x109/L (median (Q1–Q3)	12.5 (8.7–14.5)	12.5 (8.7–14.5)	0.797
3. Lymphocyte x109/L (median (Q1–Q3)	0.8 (0.6–1.2)	0.7 (0.5–0.8)	0.023
4. Monocyte x109/L (median (Q1–Q3)	0.9 (0.7–1.2)	0.9 (0.7–1.3)	0.907
5. mmol/L (median (Q1–Q3)Hb	7.6. (7–8.1)	7.7. (7–8.2)	0.652
6. Hct (%) (median (Q1–Q3)	0.35 (0.33–0.38)	0.37 (0.35–0.39)	0.152
7. MLR (median (Q1–Q3)	1.0 (0.8–1.5)	1.5 (1.1–2)	0.037
8. NLR (median (Q1–Q3)	14.6 (10–21)	10 (11–25.5)	0.231
9. Plt x103/uL (median (Q1–Q3)	209 (183–255)	205 (184–244)	0.525
Postoperative parameters (24 h):			
1. Creatinine mmol/L (median (Q1–Q3)	87 (78–105)	91 (81–114)	0.112
2. Fibrinogen mmol/L (median (Q1–Q3)	367 (293–430)	355 (315–429)	0.933
3. Troponin mcg/L (median (Q1–Q3)	1 (0.6–3.05)	0.6 (0.4–3.3)	0.247

Abbreviations: Cx—circumflex artery, Hct—hematocrit, Hb—hemoglobin, LAD—left descending artery, LIMA—left internal mammary artery, MLR—monocyte to lymphocyte ratio, NLR—neutrophil to lymphocyte ratio, RCA—right coronary artery, RIMA—right internal mammary, SVBG saphenous vein graft.

**Table 3 medicina-57-01324-t003:** Cox regression univariable analysis.

	HR	Std. Err.	Z	P > z	95% Conf. Interval
Age	0.9713	0.029	−0.97	0.334	0.92–1.03
Concomitant diseases:					
1. DM	1.13	0.52	0.27	0.786	0.46–2.78
2. HA	0.61	0.26	−1.16	0.246	0.2–1.42
3. COPD	2.86	1.46	2.07	0.039	1.06–7.77
4. Stroke	1.25	1.28	0.22	0.83	0.167–9.34
Coronary blood flow:					
1. LAD blood flow	0.99	0.02	−0.04	0.971	0.97–1.03
2. Cx blood flow	1.00	0.02	0.05	0.960	0.96–1.03
3. RCA blood flow	1.06	0.03	2.08	0.038	1.01–1.11
LVEF od admission	0.98	0.02	−1.15	0.250	0.95–1.01
LVEF on discharge	0.98	0.03	−0.77	0.442	0.92–1.04
Morphology on 1st day:					
1. WBC	1.01	0.07	0.18	0.854	0.88–1.16
2. Neutrophils	0.91	0.06	−1.46	0.144	0.81–1.03
3. Lymphocytes	0.10	0.10	−2.28	0.122	0.02–0.72
4. Monocytes	0.67	0.38	−0.7	0.483	0.22–2.06
5. NLR	1.04	0.027	1.43	0.152	0.99–1.09
6. MLR	1.28	0.28	1.13	0.259	0.83–1.98
7. MLR >1.44	3.82	1.89	2.71	0.071	1.45–10.06
8. Hb	1.01	0.33	0.02	0.981	0.53–1.90
9. Plt	0.99	0.01	−1.1	0.273	0.99–1.00

Abbreviations: COPD—chronic obstructive pulmonary disease, DM—diabetes mellitus, HA—arterial hypertension, Hct—haematocrit, Hb—haemoglobin, LVEF—left ventricle ejection fraction, LVDd—left ventricle diastolic diameter, MLR—monocyte to lymphocyte ratio, NLR—neutrophil to lymphocyte ratio, Plt—platelets, WBC—white blood count.

## Data Availability

Data supporting the reported results can be found by direct contact with the corresponding author after a justifiable explanation of requirements for three years following the publication.

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
