# Peer review of "Monocyte-to-Lymphocyte Ratio as a Predictor of Worse Long-Term Survival after Off-Pump Surgical Revascularization-Initial Report"

_medicina, 2021, doi:10.3390/medicina57121324_

Round 1

Reviewer 1 Report

I reviewed the article entitled “Monocyte-to-lymphocyte ratio as a predictor of worse survival  long-term survival after off-pump surgical revascularization –initial report.” with a great pleasure. The authors analyzed association between simply available hematologic indices and long-term mortality following off-pump coronary artery bypass grafting (OPCAB). The article is interesting and contains significant information. I have only some minor comments. My comments are below:

-Did the authors evaluate the STS and Euroscore? These two scores are the most used risk scores in patients undergoing heart surgery for in hospital mortality? Ä°f yes, was there any correlation between MLR and these parameters.

-What was the diagnosis of the patients? Chronic coronary syndrome or acute coronary syndrome? This feature may affect the prognosis of patients.

-The authors stated that Cox regression model was performed twice, once as univariable and second as a multivariable. However, only the results of univariable analysis are appeared in the manuscript. What about the results of multivariable analysis? Was this analysis performed ?

-In a previous study, many different inflammatory markers were evaluated and compared in patients with coronary artery disease. (Angiology. 2020; 71(4): 360-365”). You can discuss the following article with your findings.

-The title of article contains some mistake. I suggest authors to re-chect their title again. This section may be wrotten wrong. Please check your title in terms of mistakes.

Author Response

                                                                                 Poznan, 24th November 2021

Dear Reviewer,

I would like to thank you for your value comments regarding the manuscript. Please, find enclosed answer to your suggestions:

I reviewed the article entitled “Monocyte-to-lymphocyte ratio as a predictor of worse survival  long-term survival after off-pump surgical revascularization –initial report.” with a great pleasure. The authors analyzed association between simply available hematologic indices and long-term mortality following off-pump coronary artery bypass grafting (OPCAB). The article is interesting and contains significant information. I have only some minor comments. My comments are below:

-Did the authors evaluate the STS and Euroscore? These two scores are the most used risk scores in patients undergoing heart surgery for in hospital mortality? Ä°f yes, was there any correlation between MLR and these parameters.

The perioperative mortality risk was assessed by Euroscore II. We added the information in Table 1. We found no correlation between MLR and Euroscore II

-What was the diagnosis of the patients? Chronic coronary syndrome or acute coronary syndrome? This feature may affect the prognosis of patients.

Patients included into the study were diagnosed with chronic coronary syndrome; subject with unstable angina and acute non-ST and ST myocardial infarctions were excluded from the analysis. We added the information in the Methods section

-The authors stated that Cox regression model was performed twice, once as univariable and second as a multivariable. However, only the results of univariable analysis are appeared in the manuscript. What about the results of multivariable analysis? Was this analysis performed ?

We added lacking information on multivariable analysis :

In the multivariable analysis only MLR > 1.44 was an independent predictor of long-term mortality.

-In a previous study, many different inflammatory markers were evaluated and compared in patients with coronary artery disease. (Angiology. 2020; 71(4): 360-365”). You can discuss the following article with your findings.

We added and discussed our results with the paper recommended by the Reviewer      

-The title of article contains some mistake. I suggest authors to re-chect their title again. This section may be wrotten wrong. Please check your title in terms of mistakes.

We corrected the title

Monocyte-to-lymphocyte ratio as a predictor of worse survival long-term survival after off-pump surgical revascularization – initial report.

Kind regards

Tomasz Urbanowicz

In behalf of all-co-authors

Reviewer 2 Report

This study analysed if some accessible haematological indices, representing inflammatory processes activation, may be associated with long term survival after surgical revascularization. 

This study is very interesting and well conducted but the discussion section needs to be expanded and better focused  on the main topic of this article.  For example cite and discuss the article by Serra R, et al. Neutrophil-to-lymphocyte Ratio and Platelet-to-lymphocyte Ratio as Biomarkers for Cardiovascular Surgery Procedures: A Literature Review. Rev Recent Clin Trials. 2021;16(2):173-179.

Moreover Moderate English changes are required to get the article more readable.

Author Response

Poznan, 24th November 2021

Dear Reviewer,

I would like to thank you for your value comments regarding the manuscript. Please, find enclosed answer to your suggestions:

This study is very interesting and well conducted but the discussion section needs to be expanded and better focused  on the main topic of this article.  For example cite and discuss the article by Serra R, et al. Neutrophil-to-lymphocyte Ratio and Platelet-to-lymphocyte Ratio as Biomarkers for Cardiovascular Surgery Procedures: A Literature Review. Rev Recent Clin Trials. 2021;16(2):173-179.

Moreover Moderate English changes are required to get the article more readable.

We added and discussed our results with the paper recommended by the Reviewer. We corrected spellings.

Kind regards

Tomasz Urbanowicz

In behalf of all-co-authors

Reviewer 3 Report

The article title with this methodology is interesting . But this article is not novel. 

The study has done on small participants at a single centre with highly biased sex based selection : 16 females and 113 males. 

In reality we are not using the Monocyte-to-lymphocyte ratio as a predictor of worse survival long-term survival after off-pump surgical revascularization.

The article is very well written and the methodology is correct in my opinion. I have no technical issues with the article.
Nonetheless, although I recognize the quality of the paper, I can not find  a clinical justification to explore this topic yet another time.

But in real practice we are not recommending this Monocyte-to-lymphocyte ratio as a predictor of worse survival long-term survival after off-pump surgical revascularization by several reasons . 

The authors said that this single center retrospective study involving patients referred with diagnosis 306 of stable coronary syndrome reffered for coronary artery revascularization surgery.

Why surgically revascularization was done for stable CAD ? What is the strong reason for surgical intervention for Stable CAD. 

The Authors conclude, that MLR could be a useful tool for worse long-term prognostic factor.

Why? What is the actual usefulness of it can be regarded as worse long-term prognostic factor in every day clinical practice. When - and why - should a physician look at MLR in an Stable CAD patient ? Patients' clinical and haemodynamic status, ECG changes, troponins, CK-MB titers are not enough for a thorough analysis of the patient's condition and prognosis? Is MLR actually necessary?

The authors are advised to make a meta-analysis on that points for publication rather than single centre analysis . Overall, the analysis on correlation and causation points are essential to be discussed and presented .

Author Response

Poznan, 24th November 2021

Dear Reviewer,

I would like to thank you for your value comments regarding the manuscript. Please, find enclosed answer to your suggestions:

The article title with this methodology is interesting . But this article is not novel. 

Dear Reviewer, thank you for your valuable opinion. There are some articles concerning blood indices in coronary artery disease and we discussed them in the Discussion section. However, our paper is the first study, presenting the relation between long-term survival and OPCAB procedure, to our best knowledge.

The study has done on small participants at a single centre with highly biased sex based selection : 16 females and 113 males. 

Dear Reviewer, We added that issue in the limitation section. Differences in numbers of female and male patients result however from the fact that is a real-world study with inclusion of consecutive all-comers.

We should also point out that our previous reports based on over 2.772 pts present the similar male-to-female ratio, as common in surgical revascularization. We believe the group may be found as representative despite low-volume [Urbanowicz T, Michalak M, Olasińska-Wiśniewska A, Haneya A, Straburzyńska-Migaj E, Bociański M, Jemielity M. Gender differences in coronary artery diameters and survival results after off-pump coronary artery bypass (OPCAB) procedures. J Thorac Dis. 2021 May;13(5):2867-2873.].

In reality we are not using the Monocyte-to-lymphocyte ratio as a predictor of worse survival long-term survival after off-pump surgical revascularization.

AND : The article is very well written and the methodology is correct in my opinion. I have no technical issues with the article. 
Nonetheless, although I recognize the quality of the paper, I can not find  a clinical justification to explore this topic yet another time.

But in real practice we are not recommending this Monocyte-to-lymphocyte ratio as a predictor of worse survival long-term survival after off-pump surgical revascularization by several reasons . 

Indeed, the MLR has been rarely used. According to our clinical observations which were supported by the study results, we use is in daily practice. We strongly believe that new simple markers as MLR should be considered in clinical practice though they have not been applied so far.

This is the first study, to our best knowledge, that present the long-term results supported by profound statistical analysis in OPCAB procedures.

The authors said that this single center retrospective study involving patients referred with diagnosis 306 of stable coronary syndrome reffered for coronary artery revascularization surgery.

Why surgically revascularization was done for stable CAD ? What is the strong reason for surgical intervention for Stable CAD. 

Surgical revascularization is recommended in stable coronary artery disease according to current European Society of Cardiology guidelines published in 2018 in European Heart Journal, Volume 40, Issue 2, 07 January 2019, Pages 87–165, with class of recommendation I. “The indications for revascularization in patients with SCAD who receive guideline-recommended medical treatment are the persistence of symptoms despite medical treatment and/or the improvement of prognosis.”

Patients included into the study was diagnosed with chronic coronary syndrome; subject with unstable angina and acute non-ST and ST myocardial infarctions were excluded from the analysis after Heart Team decision.

The Authors conclude, that MLR could be a useful tool for worse long-term prognostic factor.

Why? What is the actual usefulness of it can be regarded as worse long-term prognostic factor in every day clinical practice. When - and why - should a physician look at MLR in an Stable CAD patient ? Patients' clinical and haemodynamic status, ECG changes, troponins, CK-MB titers are not enough for a thorough analysis of the patient's condition and prognosis? Is MLR actually necessary?

            Thank you for this interesting comment. In our opinion several factors may influence patients’ survival and also may influence results of single parameters (eg. chronic kidney disease may lead to chronic increase in troponin level and therefore unable easy prognostic evaluation. We believe that not a one but several prognostic factors should be taken into consideration while evaluation individual patient.

We showed that MLR can be regarded as a simple marker for worse long-term prognosis and may be applied into clinical practice. We believe, that, such simple parameter may point out those subgroups of patients which require scrutiny follow-up due to presence of excessive inflammatory reactions.

Hemodynamic changes, ECG and troponins, CK-MB titers are essential for acute syndromes diagnosis. We enclose the result from our analysis performed on larger group of patients, that is not covered by this manuscript presenting the relation between max Troponin-I and risk for long-term mortality in ROC analysis.

The authors are advised to make a meta-analysis on that points for publication rather than single centre analysis . Overall, the analysis on correlation and causation points are essential to be discussed and presented .

Thank you for this kind suggestion. We shall perform such a paper as a sequence manuscript.

Kind regards

Tomasz Urbanowicz

in behalf of all-co-authors

Round 2

Reviewer 2 Report

amended manuscript is acceptable

Author Response

Dear Reviewer,

thank you for your opinion.

I would like to thank you for your time and wiliness to review the manuscript.

Kind regards

Tomasz Urbanowicz